# Hypothesizing in the Face of the Opioid Crisis Coupling Genetic Addiction Risk Severity (GARS) Testing with Electrotherapeutic Nonopioid Modalities Such as H-Wave Could Attenuate Both Pain and Hedonic Addictive Behaviors

**DOI:** 10.3390/ijerph19010552

**Published:** 2022-01-04

**Authors:** Ashim Gupta, Abdalla Bowirrat, Luis Llanos Gomez, David Baron, Igor Elman, John Giordano, Rehan Jalali, Rajendra D. Badgaiyan, Edward J. Modestino, Mark S. Gold, Eric R. Braverman, Anish Bajaj, Kenneth Blum

**Affiliations:** 1Future Biologics, Lawrenceville, GA 30043, USA; ashim6786@gmail.com; 2Department of Molecular Biology, Adelson School of Medicine, Ariel University, Ariel 40700, Israel; bowirrat@gmail.com; 3The Kenneth Blum Behavioral & Neurogenetic Institute, Austin, TX 78701, USA; luisllnos522@gmail.com (L.L.G.); rjalali@ivitalize.com (R.J.); pathmedical@gmail.com (E.R.B.); 4Graduate College, Western University Health Sciences, Pomona, CA 91766, USA; dbaron@westernu.edu; 5Center for Pain and the Brain (P.A.I.N Group), Department of Anesthesiology, Critical Care & Pain Medicine, Boston Children’s Hospital, Boston, MA 02115, USA; dr.igorelman@gmail.com; 6Cambridge Health Alliance, Harvard Medical School, Cambridge, MA 02139, USA; 7South Beach Detox & Treatment Center, North Miami Beach, FL 33169, USA; michg8@hotmail.com; 8Department of Precision Behavioral Management, Geneus Health, San Antonio, TX 78249, USA; 9Department of Psychiatry, South Texas Veteran Health Care System, Audie L. Murphy Memorial VA Hospital, Long School of Medicine, University of Texas Medical Center, San Antonio, TX 78229, USA; badgaiyan@gmail.com; 10Department of Psychology, Curry College, Milton, MA 02186, USA; edward.modestino@gmail.com; 11Department of Psychiatry, Washington University School of Medicine, St. Louis, MO 63110, USA; drmarkgold@gmail.com; 12Bajaj Chiropractic, New York, NY 10010, USA; anish@bajajchiorpractic.com; 13Institute of Psychology, ELTE Eötvös Loránd University, Egyetem tér 1-3, 1053 Budapest, Hungary; 14Department of Psychiatry, School of Medicine, University of Vermont, Burlington, VT 05405, USA; 15Centre for Genomics and Applied Gene Technology, Institute of Integrative Omics and Applied Biotechnology, Nonakuri, Purba Medinipur 721172, West Bengal, India; 16Department of Psychiatry, Wright State University Boonshoft School of Medicine and Dayton VA Medical Centre, Dayton, OH 45324, USA

**Keywords:** Genetic Addiction Risk Severity (GARS), H-Wave, substance use disorder, Reward Deficiency Syndrome, RDS, hypodopaminism

## Abstract

In the United States, amid the opioid overdose epidemic, nonaddicting/nonpharmacological proven strategies are available to treat pain and manage chronic pain effectively without opioids. Evidence supporting the long-term use of opioids for pain is lacking, as is the will to alter the drug-embracing culture in American chronic pain management. Some pain clinicians seem to prefer classical analgesic agents that promote unwanted tolerance to analgesics and subsequent biological induction of the “addictive brain”. Reward genes play a vital part in modulation of nociception and adaptations in the dopaminergic circuitry. They may affect various sensory and affective components of the chronic pain syndromes. The Genetic Addiction Risk Severity (GARS) test coupled with the H-Wave at entry in pain clinics could attenuate pain and help prevent addiction. The GARS test results identify high-risk for both drug and alcohol, and H-Wave can be initiated to treat pain instead of opioids. The utilization of H-Wave to aid in pain reduction and mitigation of hedonic addictive behaviors is recommended, notwithstanding required randomized control studies. This frontline approach would reduce the possibility of long-term neurobiological deficits and fatalities associated with potent opioid analgesics.

## 1. Introduction

### 1.1. The Purpose

In the United States, amid the opioid overdose epidemic, nonaddicting/nonpharmacological proven strategies are available to treat pain and manage chronic pain effectively without opioids. Evidence supporting the long-term use of opioids for pain is lacking, as is the will to alter the drug-embracing culture in American chronic pain management. Some pain clinicians seem to prefer classical analgesic agents that promote unwanted tolerance to analgesics and subsequent biological induction of the “addictive brain”. Reward genes play a vital part in modulation of nociception and adaptations in the dopaminergic circuitry. The purpose of this hypothesis article argues for the novel idea that in the face of the opioid crisis, Genetic Addiction Risk Severity (GARS) testing with electrotherapeutic nonopioid modalities such as H-Wave could attenuate both pain and hedonic addictive behavior overdoses.

### 1.2. The Opioid Crisis

Noncancerous pain treatment is challenging for primary care medicine. The USA has faced an iatrogenically induced opiate/opioid epidemic that has killed thousands, with as many as 130 dying daily from a narcotic overdose [1,2]. While some argue that big pharma was not the culprit, we fervently disagree with this retort. The driver in the surge in drug overdose mortality rates has been the greater use of prescription opioid analgesics. Unintentional drug overdose deaths increased in 2007 to one every 19 min. Although initially more overdose deaths involved opioid analgesics than heroin and cocaine combined [3,4], the current accessibility of inexpensive street opiates has increased dependence on heroin [5,6,7]. A National Institute of Health (NIH) survey estimated that by 2014, 25.3 million adults suffered with pain every day for the previous three months. In 2016–2017, several thousand people died from opioid/opiate overdose, particularly with the synthetic opioid fentanyl. Fentanyl is 50 times more potent than prescription opioids. In 2016, to mitigate this rising threat to public safety, new guidelines for prescribing opioids to patients suffering with chronic pain were released by the Center for Disease Control (CDC). In 2017, morphine milligram equivalents dropped by 29%, although more than 64,000 people still succumbed to narcotics overdose, resulting in a reduction in the national life expectancy. According to the National Institute on Drug Abuse (NIDA), about 116 million Americans suffer from chronic pain currently. People who suffer from chronic pain are also more prone to having worse overall mental and physical health conditions. Owing to the contribution of big pharmaceutical industries in promoting opioid usage and subsequent addiction, the estimate is that personally, the David Sackler family will pay USD 8.3 billion in fines over ten years without any criminal charges. In 2021, the CDC estimates that 91,000 people died from opioid-related overdoses.

### 1.3. Pain Estimates

Once every 14 min, 150 million people suffer from pain conditions. About 300 million narcotic prescriptions are filled per year with costs in USD in the hundreds of billions. Some of these people die from a prescription overdose. Pain experts intend to offer needed help to pain patients. It is recognized that consumption of powerful narcotics to alleviate pain can cause high tolerance and severe withdrawal symptoms in a fairly short duration [8]. A current site that describes the impact of chronic pain in the USA can be found at https://www.cdc.gov/mmwr/volumes/67/wr/mm6736a2.htm, accessed on 31 December 2021. “Reward Deficiency Syndrome” (RDS) [9] is a genetically based hypodopaminergia known to affect about one-third of people in the United States [10]. It is understood that while a few people can tolerate powerful narcotics, and no longer want opioids after being treated for pain even after withdrawal, others, because of genetic and epigenetic insults, become enthralled with addictive-like behaviors after the pain is alleviated [11]. It is noteworthy that our group recently reported on a study utilizing the Genetic Addiction Risk Severity (GARS) test showing a high drug and alcohol risk in probands attending multipain clinics chronically prescribed opioids. In chronic pain conditions, their continued requirement for powerful narcotics may depend upon the genetic antecedents [12].

### 1.4. Why GARS

The double-edged sword for pain experts, on one hand, is that their patients may be dishonest about their actual pain level or sensitivity owing to being stuck within the “addictive process”, perhaps associated with polymorphisms of their genes associated with the reward circuitry. In contrast, patients need potent narcotics to circumvent disruptive pain-related symptoms. The problem is to establish a way to distinguish between these two types of patients at beginning of their treatment. Genetic testing may provide the solution to this problem. Though this sounds simple, and we will describe the concept in more detail, we must contemplate that our DNA may predispose to addictive-like behaviors, the environment, or particularly epigenetic processes impacting expression of genes [13]. Currently, there are at least 48 reviews and original studies on GARS, per a PubMed search conducted on 12 December 2021. Unfortunately, these articles are primarily from our group, however, we encourage others to independently confirm these early studies [9].

Nevertheless, in today’s world, with numerous people dying from legal and illegal narcotics, state laws, government organizations, and “big pharma” make it very difficult to continue treating chronic pain victims during this opioid crisis [14]. It is likely that knowing a patient’s GARS result can help provide better care by offering an in-depth view of a patient’s addiction risk and eliminating presumptions related to becoming addicted.

It is rather strange to criticize the pain expert for assisting relieve pain, and in doing so, for being accountable for the unwitting individual’s so-called “bad” behavior. In addition to this dilemma of pain experts treating patients with both acute and chronic pain, this perspective article will try to shed light on evidence-based genetic guidance to help the pain experts to circumvent guessing with “Precision Addiction Management”. Therefore, after required randomized control (RCT) studies, the hypothesis is that to help those patients who show addiction liability/vulnerability, as measured by the GARS test, pain reduction utilizing H-wave therapy without addicting analgesics will be a laudable goal.

### 1.5. The Case for Electrotherapy for Pain

Iatrogenic prescription drug abuse is the swiftest rising drug problem in the United States. The two main populations in the US at risk for prescription drug overdose are roughly the 9 million people who report long-term medical opioid use and around 5 million people who report nonmedical use. The 20% of patients prescribed high daily doses and receiving care from several physicians account for nearly 80% of overdoses and are more prone to give drugs to others who consume them without prescription [15].

Additionally, the main pain pathways that arise from the dorsal horn of the spinal cord to the medulla, and numerous genes and their polymorphisms that inhabit the mesolimbic reward center of the brain have a function in the control of pain sensitivity and tolerance [16,17,18].

Identifying the reward genes and their polymorphisms may provide distinctive therapeutic targets for non-narcotic pharmacogenomic solutions to cure pain. The GARS test [19] can recognize patients with a susceptibility to addiction in the initial stages of treatment; for example, reward genes’ alleles such as DRD2 A1 and the G allele of the Mu Opioid Receptor are associated with a risk for narcotic addiction. These are the patients who will require a nonaddictive alternative pain treatment. The electrotherapeutic H-Wave^®^ device (Electronic Waveform Lab Inc., Huntington Beach, CA, USA) is one such alternative [20].

### 1.6. The Characteristics of H-Wave Electrotherapy

The physiological mechanisms of action of the H-Wave device stimulation (HWDS) examined in animals reduced edema owing to the stimulation of smooth muscle fibers within the lymphatic vessels [21]. Additionally, using HWDS aids tissue healing by the induction of nitric oxide (NO)-dependent microcirculation augmentation and angiogenesis [22].

The characteristics of HWDS include:Contraction of smooth and skeletal muscle (red, slow-twitch) fibers through low-frequency (1–2 Hz) stimulation results in tissue loading whilst maintaining the low muscle force tension characteristics, thus being nontetanizing and nonfatiguing;Arteriolar vasodilation associated with HWDS is attributed to a NO mechanism, as shown in rat studies;Increased angiogenesis demonstrated via bromouridine staining in repetitive stimulation in rats;HWDS specifically and directly stimulates the smooth muscle fibers within the lymphatic vessels, finally resulting in fluid shifts and decreasing edema and protein clearance.

There is a necessity for nonpharmacological substitutes to treat pain during the opioid crisis. The published peer-reviewed literature related to the positive effects of H-Wave includes a total of over 18 publications. These original articles, reviews, and abstracts illustrate an important, evidence-based series demonstrating noteworthy pain relief and mechanisms of action [19,20,21,22,23,24,25,26,27,28,29]. Additional studies show a vital role for electrotherapy for pain [29,30,31].

Markedly, in the face of our most awful drug crisis, with many lives lost daily, the whole pain community should adopt an alternative to potent pain medications.

### 1.7. Potential Mechanism of H-Wave Therapy

During the last two decades, researchers have been progressively interested in controlling pain and restoring function via electrical stimulation. One of the focus areas is use of H-Wave device [32].

The purpose of the HWDS is to reduce chronic pain and inflammation. Four mechanisms achieve this aim: first, through interstitial fluid shifts produced at very low frequencies (1–2 Hz) by direct stimulation of small-diameter skeletal muscle fibers and smooth muscles of the lymphatic system. HWDS causes long rhythmical contractions of these specific muscle types, reducing the accumulation of proteins associated with inflammation, an important component of pain, and related disability in trauma or chronic injury patients [29,33];Second, the H-Wave device also produces profound anesthetic/analgesic effects when used at high frequencies (60 Hz) by affecting the function of the sodium pump within the nerve [34];Third, animal research has demonstrated that skeletal muscle stimulation by the H-Wave device leads to a significant increase in microcirculation, which was NO-dependent [22,35].Fourth, repetitive HWDS to rat hind limbs produced a profound and swift increase in blood flow as a function of observed angiogenesis [23,28,36,37]. These factors obviate the likelihood that the repetitive HWDS decreases inflammation and supports faster healing and improved recovery due to reducing protein buildup in postoperative conditions such as rotator cuff reconstruction.

Blum et al. published a meta-analysis where they systematically reviewed the efficacy and safety of the HWDS as a nonpharmacological analgesic treatment in chronic soft tissue inflammation and neuropathic pain. The analysis included five studies related to pain relief, pain medication reduction, and increased function achieved with the H-Wave device. Data were analyzed using the random-effects model, including adjustment to evaluate variability, study size, and bias in effect size [21]. The meta-analysis used data from a total of 6535 participants [21,38,39,40]. In this specific meta-analysis, though the findings indicate a moderate-to-strong effect of the H-Wave device in providing pain relief, a reduced necessity for pain medication, and enhanced function, we suggest additional studies. The most robust effect was observed for increased function, suggesting that the H-Wave device may help a faster return to work and other related daily activities [41].

### 1.8. Rationale for Pain Reduction

Pain may be undertreated, contributing to agony, as stated by the World Health Organization (WHO). Pain may also be overtreated, unintentionally leading to drug addiction, drug diversion, and even death. Thus, “*primum non-nocere*—*first, do no harm*”—is not easily attained, in the pharmacological pain treatment, especially in chronic pain. In 2008, Henn et al. [42] reported in a prospective study involving 125 patients with Workers’ Compensation claims worse outcomes, even after controlling for confounding factors including, age, work demands, lower marital rates, education levels, preoperative expectations, compared to non-Workers’ Compensation patients. This study by Henn et al. delivers proof that the presence of a Workers’ Compensation claim portends a less robust outcome following a rotator cuff repair and stimulated interest in assessing HWDS to improve outcomes.

Additionally, according to Kasten et al. [43], even with today’s ultra- technical elegance, shoulder surgery can cause significant pain. Data from randomized controlled trials (RCTs) resulted in recommendations for local or regional anesthesia for analgesia during and after surgery of the upper extremity. This treatment entails potent addictive opioids and nonsteroidal anti-inflammatory drugs (NSAIDs) in a multimodal analgesia method. Moreover, according to a meta-analysis of RCTs, since the pain is profound, an interscalene nerve block is suggested for analgesia during and after surgery of the shoulder. Other recommendations include physiotherapy postoperatively. Interestingly, while the use of arthroscopic procedures for most knee conditions lead to comparatively mild and controlled pain, it is known that arthroscopic procedures for rotator cuff repair and reconstruction can lead to more significant pain for the patient’s undergoing recovery, and thus remains a greater challenge.

The introduction of pain pumps at first was met with enthusiasm by several shoulder surgeons but has resulted in serious complications involving chondrolysis. Various studies utilizing bovine and rabbit cartilage suggested that there is significant chondrotoxicity from bupivacaine, a local anesthetic frequently used in pain pumps [44].

### 1.9. Linking GARS Testing to Medical Necessity for Nonaddicting H-Wave Therapy

Millions of Americans suffer through pain daily. In 2017, opioid overdose took 64,000 lives, increasing to 84,000 lives in 2020 and 91,000 in 2021, resulting in decreased national life expectancy. Long-term opioid usage results in dependency, drug tolerance, neuroadaptation, hyperalgesia, potential addictive behaviors, or RDS caused by hypodopaminergia [45]. Table 1 displays the genes and associated risk alleles measured in the GARS test.

Evaluation of pain patients with the GARS test and the Addiction Severity Index (ASI-Media Version V) showed that GARS scores equal to or greater than 4 and 7 alleles considerably predicted drug and alcohol severity, respectively [41]. In a recent study by Moran et al. [12], we used RT-PCR for SNP genotyping and multiplex PCR/capillary electrophoresis for fragment analysis of the role of 11 alleles in a 10 reward-gene panel, displaying the activity of brain reward circuitry in 121 chronic opioid users. The study comprised 55 males and 66 females, with an average age of 54 and 53 years, respectively. The patients included Caucasians, African Americans, Hispanics, and Asians. Inclusion criteria required that the Morphine Milligram Equivalent (MME) was 30–600 mg/day for males and 20–180 mg/day for females, for treatment of chronic pain over 12 months. In total, 96% carried four or more risk alleles, and 73% carried seven or more risk alleles, implying a high predictive risk for opioid and alcohol dependence, respectively. These data suggested that chronic, licit prescribed opioid users going to a pain clinic possess a high genetic risk for drug and alcohol addiction. Early recognition of genetic risk, using the GARS testing upon entry to treatment, might prevent iatrogenic induced opioid dependence. Upon entry to pain clinic, a score of above four risk alleles may provide clinicians with the medical necessity to prescribe H-Wave instead of potent addictive analgesics. Utilization of this novel approach has the clinical potential to reduce iatrogenically induced subsequent addiction liability, especially in patients characterized as high risk for all addictive or reward-deficiency addictive behaviors, both drug and nondrug [46]. Figure 1 is a schematic of the proposed alternative treatment of pain.

While it is true that the major body of the RDS concept resides in many publications on the subject from our laboratory, there is an emerging increase in independent citations that embrace this novel construct. A brief sampling of this global cited work, while mostly favorable, can be assessed by reviewing specific references [47,48,49,50,51,52,53,54,55,56,57,58,59,60,61,62,63,64,65,66,67,68,69,70,71,72,73,74,75,76,77,78,79,80,81,82,83,84,85,86,87,88,89,90,91,92,93,94,95,96,97,98,99,100,101,102,103,104,105,106,107,108,109,110,111].

## 2. Conclusions

It is important to realize that vulnerability to drug and even nondrug addictive behaviors may reside in both genetic and epigenetic antecedents and impacts, respectively. In many important definitions of addiction, the genetic role is highlighted in relation to other factors, such as the environment, life experiences, the psychology of the individual. There are a number of examples revealing that people with addiction use substances or engage in behaviors that become compulsive and often continue despite harmful consequences [https://www.asam.org/QualityScience/definition-of-addiction; https://www.apa.org/topics/substanceuse-abuse-addiction, accessed on 31 December 2021]. Of cause, there also exists the issue related to the neurobiology of spirituality and environmental epigenetic insults as well [47,48,49]. The authors are proposing a paradigm shift, whereby at entry in pain clinics, the potential coupling of the GARS test with H-Wave modality could attenuate both pain and addiction. However, while there is indeed a scientific foundation related to the efficacy and rationale to utilize H-Wave, there is a continual need to research this important DNA-guided combination approach to reduce unnecessary utilization of long-term treatment with opioids, especially in high-risk, genetically vulnerable populations. Precision addiction management uses GARS test results to treat RDS in these genetically vulnerable populations. This frontline approach will also potentially temper the possibility of long-term issues and possible fatalities associated with addictive opioid analgesics. This dual-modality, H-Wave, and GARS testing, if adopted to help treat pain and RDS, could indeed reduce iatrogenic opioid-induced fatalities.

## Figures and Tables

**Figure 1 ijerph-19-00552-f001:**
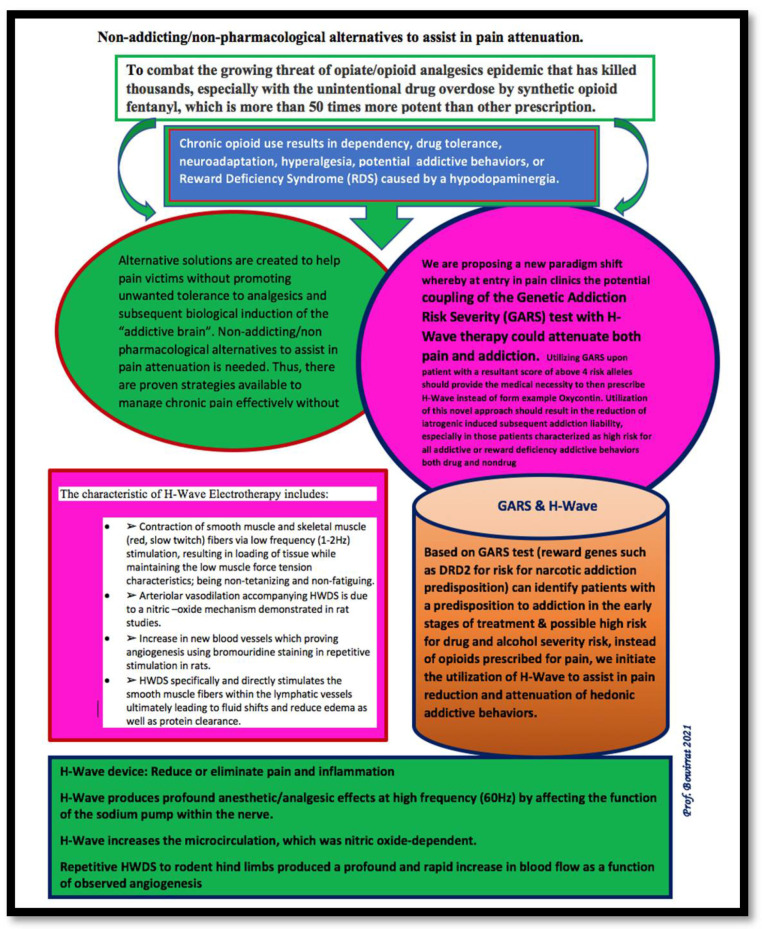
A nonaddicting/nonpharmacologic alternative to aid in pain mitigation.

**Table 1 ijerph-19-00552-t001:** Representation of the GARS SNPs and VNTRs (snapshot).

Gene	Polymorphism	Location	Risk Allele(s)
Dopamine D1 Receptor DRD1	Rs4532 SNP	Chr 5	A
Dopamine D2 Receptor DRD2	Rs1800497 SNP	Chr11	A
Dopamine D3 Receptor DRD3	Rs6280 SNP	Chr 3	C
Dopamine D4 Receptor DRD4	Rs1800955 SNP	Chr 11	C
	48 bases repeat VNTR	Chr 11, Exon 3	7R, 8R, 9R, 10R, 11R
Catechol-O-Methyltransferase COMT	Rs4680 SNP	Chr 22	G
Mu-Opioid Receptor OPRM1	Rs1799971 SNP	Chr 6	G
Dopamine Active Transporter DAT1	40 bases repeat VNTR	Chr 5, Exon 15	3R, 4R, 5R, 6R, 7R, 8R
Monoamine Oxidase A MAOA	30 bases repeat VNTR	Chr X, Promoter	3.5R, 4R
Serotonin Transporter SLC6A4 (5HTTLPR)	43 bases repeat INDEL/VNTR plus rs25531 SNP	Chr 17	LG, S
GABA (A) Receptor, Alpha 3 GABRB3	CA-Repeat DNR	Chr 15 (downstream)	181
Abbreviations: Single Nucleotide Polymorphisms (SNP), Variable Number Tandem Repeats (VNTR).r 17			

## Data Availability

Not applicable.

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
