# Peer review of "Hypothesizing in the Face of the Opioid Crisis Coupling Genetic Addiction Risk Severity (GARS) Testing with Electrotherapeutic Nonopioid Modalities Such as H-Wave Could Attenuate Both Pain and Hedonic Addictive Behaviors"

_ijerph, 2022, doi:10.3390/ijerph19010552_

Round 1

Reviewer 1 Report

Thank you for the opportunity to review this study entitled “Hypothesizing in the face of the opioid crisis coupling Genetic Addiction Risk Severity (GARS) testing with electro-therapeutic non-opioid modalities like H-Wave could attenuate both pain and hedonic addictive behaviors” (ijerph-1488470).

The manuscript presented a valuable perspective, concerning the use of opioids in the treatment of pain and the risk of developing an addiction, highlighting the importance to consider the Genetic Addiction Risk Severity (GARS) test coupled with the H-Wave at entry in pain clinics to attenuate pain and help prevent addiction.

In my opinion, the research topic is relevant, and the paper is interesting. Parallelly, there are some minor issues that need to be addressed before the paper will be suitable for publication:

  • Lines 83-85. Please, clarify the bibliographic reference relating to these data.
  • Lines 159- 168. This part looks a bit chaotic. The list of the journals (some cited in abbreviated form, others in the extended one) makes reading more strenuous. Perhaps this section could be streamlined by simply focusing on its core aspect: the results of these studies.
  • Lines 175-187. I suggest using the bulleted list to make this section of the text easier to read.
  • Line 201. Please, provide a clear bibliographic reference for this sentence.
  • This paper focuses on two important key aspects, related to each other: 1) the treatment of pain; 2) vulnerability to addiction. In reference to the latter, the genetic perspective is certainly very important. However, it is not the only one. In fact, in many important definitions of addiction, the genetic role is highlighted in relation to other factors, such as the environment, life experiences, the psychology of the individual. Here are two examples:
    • Addiction is a treatable, chronic medical disease involving complex interactions among brain circuits, genetics, the environment, and an individual’s life experiences. People with addiction use substances or engage in behaviors that become compulsive and often continue despite harmful consequences” (ASAM, Retrived at: https://www.asam.org/Quality-Science/definition-of-addiction)
    • Addiction is a chronic disorder with biological, psychological, social and environmental factors influencing its development and maintenance.”  (APA, Retrieved at: https://www.apa.org/topics/substance-use-abuse-addiction)

Therefore, the paper offers an interesting and really important perspective that focuses on the genetic aspect. However, since addiction is also defined in psychological, social and environmental terms, in the introductory sections or in the conclusion it is important to make a brief and quick mention of other vulnerability factors for addiction (Caretti et al., 2018, https://doi.org/10.3390/jcm7080194; Fouyssac et al., 2021, https://doi.org/10.1111/ejn.15087; Everitt et al., 2008; https://doi.org/10.1098/rstb.2008.0089), underlining the importance of maintaining a vision that integrates all of these elements as much as possible when dealing with this issue.

Reviewer 2 Report

Opioid misuse and addiction are a public health crisis resulting in debilitation, deaths, and significant social and economic impact. Curbing this crisis requires collaboration among academic, government, and industrial partners toward the development of effective nonaddictive pain medications, interventions for opioid overdose, and addiction treatments. Therefore, the subject matter proposed by the Authors is very topical and of great clinical importance. The solutions proposed in the manuscript seem very interesting and may significantly reduce the opioid crisis. However, I have some suggestions for Authors. In my opinion, the Authors should:

- clearly distinguish the purpose of the work as a separate point

- expand the conclusions as they stand are too vague

- expand the information on genetic addiction risk severity - insufficient in its current form

Reviewer 3 Report

In the reference list there are at least 17 self-citations out of the total 45 cited papers.  Some terminologies mentioned in this paper, such as 'reward deficiency syndrome (RDS)' or 'Genetic addiction Risk severity (GARS) test', were not backed up by any other sources besides self-citation.  Moreover, the paper provides a lot of assumptions and deductions which again were not supported by  reliable evidences.  Even when non-self-cited papers were referenced, they were sometimes too general or off-target and not supportive enough for the authors' view points (e.g.: H-wave therapy in section 1.6).
